# Internal water circulation mediated synergistic co-hydrolysis of PET/cotton textile blends in gamma-valerolactone

Shun Zhang [1], Wenhao Xu[1], Rongcheng Du[1], Lei Yan[1], Xuehui Liu[2], Shimei Xu [1] ✉ & Yu-Zhong Wang [1]

Recycling strategies for mixed plastics and textile blends currently aim for recycling only one of the components. Here, we demonstrate a water coupling strategy to co-hydrolyze polyester/cotton textile blends into polymer monomers and platform chemicals in gamma-valerolactone. The blends display a proclivity for achieving an augmented 5-hydroxymethylfurfural yield relative to the degradation of cotton alone. Controlled experiments and preliminary mechanistic studies underscore that the primary driver behind this heightened conversion rate lies in the internal water circulation. The swelling and dissolving effect of gamma-valerolactone on polyester enables a fast hydrolysis of polyester at much lower concentration of acid than the one in the traditional hydrolysis methods, effectively mitigating the excessive degradation of cotton-derived product and undesirable product formation. In addition, the system is also applicable to different kinds of blends and PET mixed plastics. This strategy develops an attractive path for managing end-of-life textiles in a sustainable and efficient way.

In the past two decades, global fiber production has nearly doubled and it is forecasted that fiber consumption will reach 160 million tons in 2050[1–3]. The current fast fashion trend not only leads to increased demand for new garments but also a significant amount of textile waste[4,5]. The impact of the fashion industry on the environment is widespread and thought-provoking, accounting for an estimated 8–10% of global $CO_2$ emissions[6]. Meanwhile, the textile wastes are predominantly subjected to landfilling or incineration at the end of life, resulting in a depletion of resources and environmental pollution[7–9], as well as the potential health threat of microfibers, which have been found in rivers, oceans, and drinking[10–12]. To tackle environmental concerns, chemcycling is regarded as a promising method to develop sustainable fashion and circular economy[13–15]. However, textiles frequently comprise diverse polymers, of which the polyester/cotton textile blends account for the vast majority[16]. In the process of

chemcycling, components other than the target components are generally considered to be unfavorable to the reaction. Separation is usually adopted prior to chemical degradation through manual or machine sorting[17–19], as well as selective dissolution by solvents (Fig. 1a)[20–22]. However, It makes the whole process time-consuming and energy-extensive. An alternative way is to directly degrade one component of textile blends while retaining the other intact (Fig. 1b). Current research efforts predominantly concentrate on converting easily treated cotton fibers into chemicals, such as sugars or furfural, using acid or ionic liquid, while keeping the polyester unchanged[23,24]. Differently, in our recent work, in-situ alkaline hydrolysis of polyester into terephthalic acid (TPA) is achieved while cotton fibers remain intact in $CH_2Cl_2$-EtOH solvents[25]. Generally, these strategies take advantage of the differences in solubility or degradation selectivity among the components of the blends to recover a specific component

[1]Collaborative Innovation Center for Eco-Friendly and Fire-Safety Polymeric Materials (MoE), State Key Laboratory of Polymer Materials Engineering, National Engineering Laboratory of Eco-Friendly Polymeric Materials (Sichuan), College of Chemistry, Sichuan University, Chengdu, China. [2]Collaborative Innovation Center for Eco-Friendly and Fire-Safety Polymeric Materials (MoE), State Key Laboratory of Polymer Materials Engineering, National Engineering Laboratory of Eco-Friendly Polymeric Materials (Sichuan), College of Architecture and Environment, Sichuan University, Chengdu, China. ✉e-mail: xushimei@scu.edu.cn

**Fig. 1 | Chemical recycling strategies of polyester/cotton blends. a** Degradation after separation strategy and **b** selective depolymerization of specific components in the traditional method. **c** Synergistic co-degradation strategy in this work.

while preserving the integrity of the others. However, additional steps will be required to recover the other components, which results in additional energy costs.

In addition, in the process of recycling the mixture, the by-products produced by one polymer may have unexpected effects on the transformation of another polymer. Recently, Ma[26] reported a method of in-situ utilization of chlorine released by PVC, which successfully converted PET into terephthalic acid and 1,2- dichloroethane with high yield. Different from the previous experiences where chlorine in PVC was usually considered to be harmful to the transformation of other polymers due to catalyst poisoning, the above work demonstrated that chlorine can also play an active role in the transformation of the mixture into valuable products. However, the simultaneous transformation of textile blends into high-valued chemicals is more intriguing but very challenging due to different, even conflicting, degradation needs among different components. We found that 5-hydroxymethylfurfural (HMF) is an important bio-based platform chemical for the production of renewable monomers and bio-fuels, which can be produced by acidolysis of cotton.

However, the generation of HMF involves a dehydration step, and an increase in water content may impede the favorable progression of the reaction. The yield of HMF recovered from cotton degradation is often less than 15%[3,27,28]. Considering that the acidic hydrolysis of both polyester and cotton is a water-consuming process, it is promising to achieve synergistic co-degradation of polyester/cotton blends by utilizing the byproduct water from the HMF dehydration step to meet the water consumption requirement for the hydrolysis of polyester and cotton. However, due to the differences between the two polymers, there are two main issues that need to be addressed in the acidic co-hydrolysis of polyester/cotton blends into TPA and HMF. One is to avoid the formation of undesirable byproducts from cotton in high concentrations of acid, such as 80 wt% para-toluenesulfonic acid (TsOH)[29]. The high acid concentration is necessary for polyester hydrolysis due to the heterogeneous nature of the reaction but may lead to excessive degradation of cotton, forming undesirable byproducts. The other is to strike a balance between HMF transformation and polyester/cotton hydrolysis through internal water circulation.

## Table 1 | Degradation of polyester and cotton [a]

| Entry | Substrate | Solvent | Yield (%) | |
|---|---|---|---|---|
| | | | TPA [c] | HMF [d] |
| 1 | polyester | GVL | 96.3 ± 1.6 | / |
| 2 | polyester | H$_2$O | <1 | / |
| 3 | cotton | GVL | / | 19.7 ± 1.1 |
| 4 | cotton | H$_2$O | / | 4.34 ± 1.1 |
| 5 | polyester/cotton | GVL | 95.8 ± 1.9 | 24.1 ± 1.0 |
| 6 | polyester/cotton | H$_2$O | <1 | 4.73 ± 0.9 |
| 7 | polyester/cotton | GVL/ H$_2$O [b] | 28.3 ± 1.4 | 7.38 ± 1.3 |

Polyester and cotton were cut from textiles; [a] Reaction under 170 °C for 60 min with 7.5 wt% TsOH, the solvent is 20 mL. Unless otherwise specified, the polyester/cotton blends utilized in this work is 1 g (with a polyester to cotton mass ratio of 8:2). In experiments involving the individual use of cotton or polyester, the masses of polyester and cotton are 0.8 g and 0.2 g, respectively. [b] The volume ratio of GVL to H$_2$O is 5:5; [c] Yield determined by weight; [d] Yield determined by NMR and HPLC.

In this work, we show a water coupling strategy to co-hydrolyze polyester/cotton textile blends into polymer monomers and platform chemicals. We consider gamma-valerolactone (GVL) as the reaction solvent, utilizing its excellent swelling and solubility properties on polyester to enhance interfacial mass transfer (Fig. 1c). This, in turn, enabled the degradation of polyester at low acid concentration. Furthermore, we found that internal water circulation meets the needs of the synergistic conversion of blends. The water generated from the production of HMF is utilized to participate in the degradation of polyester, while the water consumed by the depolymerization of polyester promotes the conversion of HMF. This strategy opens up an attractive path for managing commercial waste textiles and lays the groundwork for energy-efficient chemcycling for diverse waste blends.

## Results

### Degradation of polyester/cotton blends

Here, we employed an eco-friendly solvent GVL to simultaneously degrade polyester and cotton in polyester/cotton blends into high-value products under a concentration of TsOH as low as 7.5% at 170 °C for 1 h (Fig. 1c). The concentration of TsOH is only 10% of the one in the traditional hydrolysis of polyester[29]. Polyester was completely depolymerized to obtain TPA with high yield (>95%) and high purity (>99%) (Supplementary Fig. 1-3), which is comparable to that by two-step alkaline depolymerization of polyester[30]. Meanwhile, the cotton was degraded into versatile platform compound HMF with yield of 25% (Supplementary Fig. 4-5), which is almost twice as much as that in traditional degradation method of cotton[31]. Furthermore, no additional water was added in the whole process, indicating a good matching between water production and consumption.

### Internal water circulation of polyester/cotton blends degradation

As observed in the experiment above, the yield of HMF from polyester/cotton blends is higher than that from pure cotton (Table 1, **entry 5 and entry 3**). We speculate that polyester plays a positive role in the conversion process of cotton. It was further confirmed by the fact that the yield of HMF exhibited a decreasing trend as the polyester content in the blends gradually decreased (Fig. 2a). Transformation of HMF from cotton is a dehydration process, and prompt removal of water will facilitate the further reaction. Through real-time monitoring of the water content in the degradation solution, it was observed that the water content initially decreased and then remained stable when the

sole cotton was degraded, but a continuous decrease for the polyester/cotton blends (Fig. 2b). This phenomenon suggests that an equilibrium is reached between the water consumption from cotton hydrolysis and water production from fructose dehydration in the case of cotton. However, the water content after reaching equilibrium is not conducive to HMF formation. With the addition of polyester, water consumption increased. The reaction equilibrium above was broken, which promoted HMF formation. As a result, the introduction of external water is not conducive to the forward progress of the generation of HMF (Fig. 2c). When GVL was replaced by water, the yield of HMF was only about 4%, which is 5 times lower than the one in GVL no matter whether polyester/cotton blends or pure cotton (Table 1, **entry 6 and entry 4**). Besides, the increase of water content decreased the degradation rate of polyester accordingly due to weakened swelling/solubility of PET in GVL (Fig. 2d). This also explained why polyester fiber hardly depolymerized at the same temperature and time in aqueous solution (Table 1, entry 2, Supplementary Fig. 6).

To further elucidate the mechanism of water circulation, the degradation rate of polyester/cotton blends and HMF yield at different reaction times was investigated (Supplementary Fig. 7-8, Supplementary Table 1). Within the initial 10 min of the reaction, almost no polyester depolymerization and HMF formation were observed, except that only a minor amount of cotton hydrolysis occurred. It suggested that the glucose accumulated in the stage. Subsequently, the glucose underwent isomerization and dehydration procedures to generate HMF (Supplementary Fig. 9-10). Meanwhile, the degradation rate of both polyester and cotton in the blends gradually rose. Compared to sole cotton, the degradation rate of cotton in the blends showed minimal variation over time, demonstrating that the presence of polyester did not significantly impact cotton degradation. However, HMF yield was notably higher than that in sole cotton accordingly. Especially, there was a sharp acceleration in the polyester degradation in a period of 40 and 60 minutes, which corresponded to a rapid decrease of water content in the degradation solution. Accordingly, the HMF yield also increased to 25%, indicating that the water consumption in polyester depolymerization facilitated the dehydration of fructose to produce HMF.

### Solvent effect on the degradation process

To further investigate the significant differences in the degradation of polyester and cotton in GVL and water, XRD spectra were performed on the residues of cotton reacted in GVL and water at different times, respectively. The crystalline peak of cotton at 22.6° significantly decreased in GVL while showing slight changes in water with increasing reaction time up to 40 min (Supplementary Fig. 11). The crystallinity of cotton in GVL showed a trend of initially increasing and then decreasing with time (Fig. 3a). This was because that the degradation in the initial stage of reaction were inclined to take place in the amorphous region of cotton, leading to an increase in its crystallinity of residual cotton. However, with further extending of reaction time, the crystalline region was destructed, resulting in a decrease in crystallinity. At 30-40 min, the crystallinity of cotton decreased from 76% to 37%, indicating a significant conversion at this stage. In contrast, the crystallinity of cotton in water showed a steady upward trend, which was because the degradation rate of cotton in water was relatively slow, and the crystalline region was destroyed in a limited degree. A similar phenomenon was observed in the reaction of polyester, where the peak intensity of polyester in GVL changed more significantly than in water (Supplementary Fig. 12). The crystallinity of the residual solid material after different reaction time in water remained stable at around 43% (Fig. 3b), which was consistent with the previous observation that polyester was hardly degraded in water (Table 1, **entry 2**). However, in GVL, the crystallinity of polyester showed a trend of

 

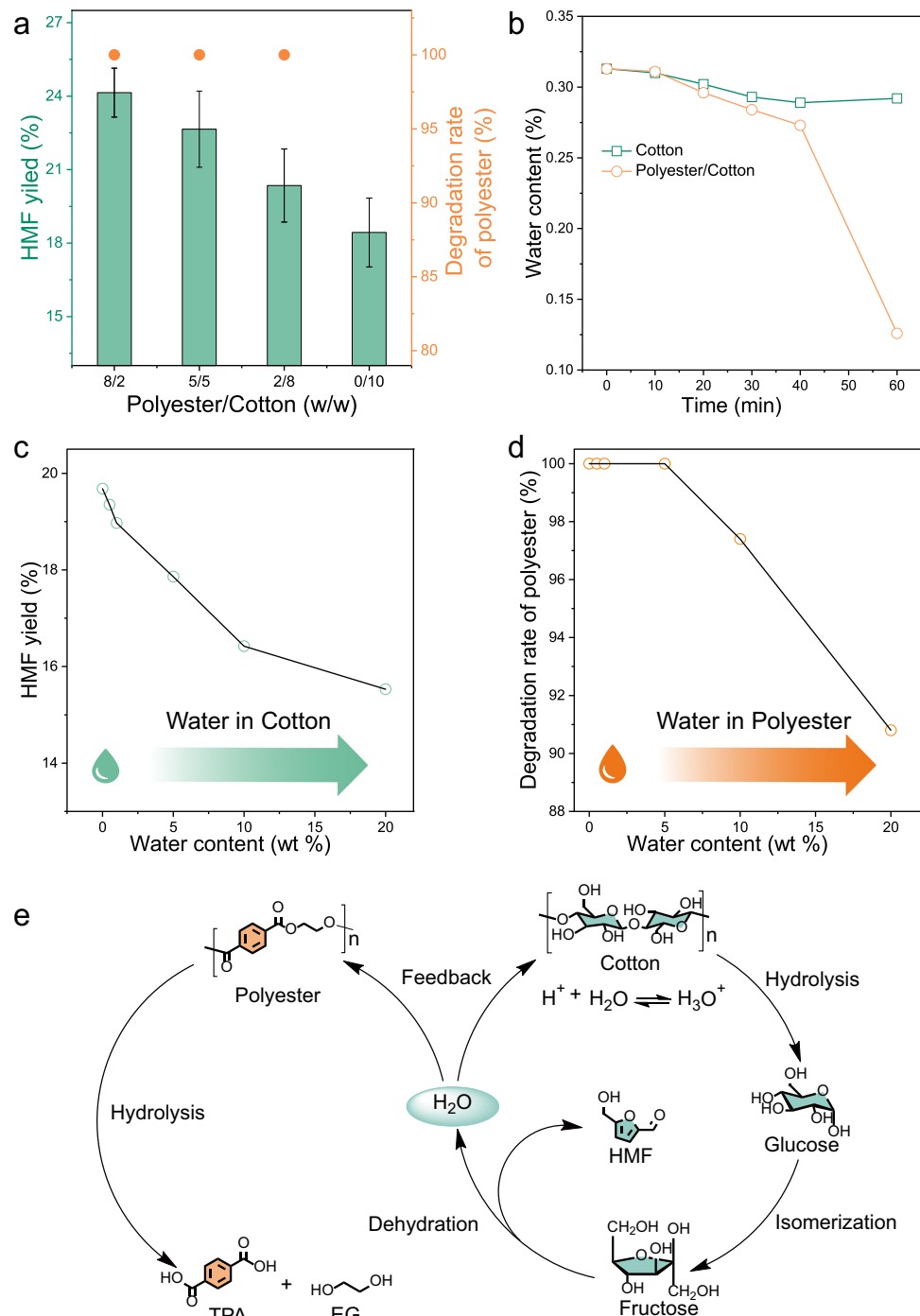

**Fig. 2 | Mechanism of internal circulation of water. a** The effects of polyester/cotton blends ratios on HMF yield distribution. The data points are from N = 3 independent experiments. **b** Water content in degradation solution of polyester/cotton blends or cotton at different reaction time (the water content in the initial solution is 0.31%); (**c**) Effect of different contents of water in GVL on HMF yield from cotton and (**d**) on the degradation rate of polyester. **e** Internal water circulation in the degradation process of polyester/cotton blends. Reaction under 170 °C with 7.5 wt% TsOH, the GVL is 20 mL.

initially decreasing and then increasing with the extending of hydrolysis time. It is contrary to the cotton. At the lower temperatures, polyester either experienced swelling or partial dissolution in GVL. However, at 170 °C, a transparent solution was observed, indicating the complete dissolution of polyester (Fig. 3c). Upon cooling, a white solid precipitated from the solution, signifying the ability of polyester to dissolve in GVL, thus promoting its degradation at this reaction temperature. However, in water, polyester remains basically unchanged. The FT-IR spectra of polyester treated with GVL for different durations indicated that the ester groups of polyester remained intact

during the process, signifying that there were no bonds broken apart from physical fragmentation (Supplementary Fig. 13). We further examined the structural alterations in polyester during the dissolution process at 170 °C in GVL (Fig. 3d). After a 0.5 h treatment, a white solid was obtained with original flake-like structure upon cooling the solution. After 1 h, almost all of the polyester transformed into a paste-like solid, and after 3 h, no polyester residue remained in the solution. It explains a sudden increase of the degradation rate is due to the dissolution of polyester in GVL. However, the morphology of polyester fiber remained almost unchanged in water after 3 h treatment. Due to

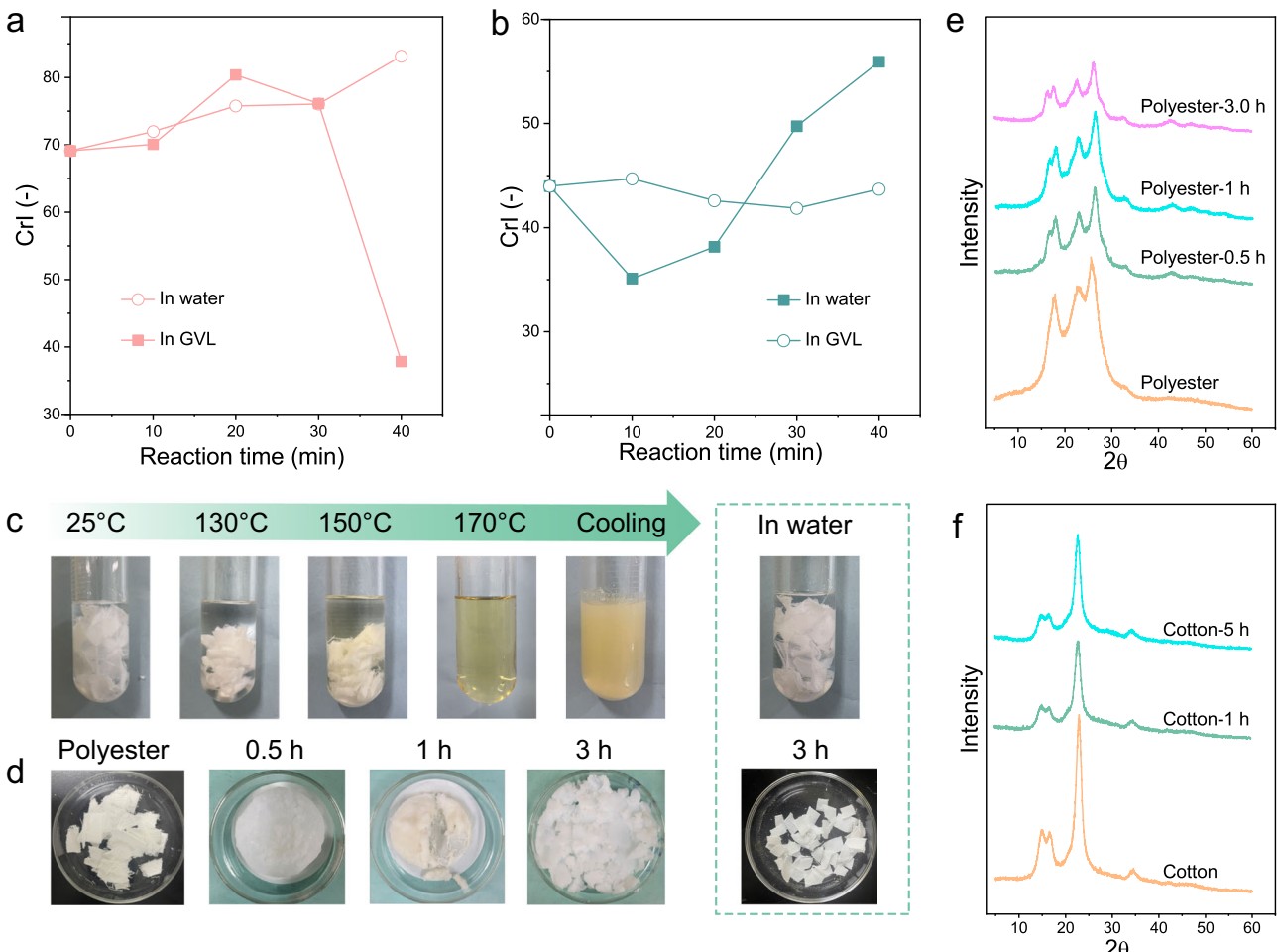

**Fig. 3 | Monitoring the degradation process of polyester/cotton blends. a** XRD spectra of cotton in GVL and water at different reaction time, **b** XRD spectra of polyester in GVL and water at different reaction time. Reaction conditions: 7.5% TsOH, 170 °C, 20 mL solvent. Polyester treated by GVL and water at different temperatures for 3 h (**c**) and different time at 170 °C (**d**); XRD spectra of polyester (**e**) and cotton (**f**) treated with GVL at different time.

the disintegration of the physical structure, polyester experienced alterations in its crystallinity during the dissolution process. The crystallinity of initial polyester was 42.2% and then decreased to 30.6% after treatment with GVL for 3 h (Fig. 3e).

It indicated that GVL not only efficiently dissolves polyester to increase the reaction surface area but also reduces its crystallinity during the degradation process. However, GVL did not significantly affect the structure of cotton. Even after treating cotton with GVL for 5 h, its network structure remained intact (Fig. 3f and Supplementary Fig. 14). The above swelling/degradation process of polyester/cotton blends in GVL may have undergone the following stages: Initially, glucose accumulates during the decomposition of cotton. At this stage, PET is experiencing a swelling-dissolution process and is not yet involved in the water recycling process. Over time, glucose is isomerized into fructose and then fructose undergoes dehydration to form HMF. The water generates in this process, which is fed back into the hydrolysis process of polyester/cotton blends, facilitating internal water circulation and continuously driving the reaction forward. The entire reaction pathway is depicted in Fig. 2e.

The abrupt increase of degradation rate of polyester from 150 °C to 170 °C further confirmed that the dissolution of PET in GVL accelerated the degradation (Fig. 4a). Below 20% degradation rate for polyester was observed while more than 90% for cotton at 150 °C. Further, by increasing the temperature to 170 °C, 100% of the degradation rate of both polyester and cotton was achieved. Even if we conducted an extensive analysis of the polyester degradation rate

within the temperature range of 150-170 °C, the polyester degradation rate was still less than 50% below 165 °C (Fig. 4b). The kinetic analysis reveals that, in the temperature range of 110-150 °C, the reaction rate constant for polyester experiences a slight increase, suggesting a low reactivity within this temperature interval. The rate constant increased by a factor of 20 at 170 °C compared to the one at 150 °C (Fig. 4c). This substantial increase can be attributed to the effective mass transfer due to the complete dissolution of polyester in GVL at 170 °C. This rate constant in GVL is 100 times as much as the one in aqueous solution (Fig. 4d). As a result, only one-tenth of the acid dosage was required compared to the one in the traditional polyester hydrolysis method[29]. It greatly suppressed the possible side reaction in acidolysis of cotton (Supplementary Fig. 15-16). These findings confirmed that GVL can effectively accelerate the degradation rate in low acid concentrations. To verify the stability of GVL in the reaction, we employed the structural unit of polyester, bis(hydroxyethyl)terephthalate (BHET), as a model compound to simulate the reactivity difference. It was calculated that the Gibbs free energy of BHET and GVL was 15 kJ/mol and 31.5 kJ/mol, respectively (Fig. 4e), aligning with the experimental observation that only polyester can be degraded while GVL remains intact. This finding further supports the use of GVL as a suitable reaction solvent for polyester.

TPA as an acidic substance might affect the cotton conversion. No degradation of cotton was observed in the absence of an acid catalyst. However, when TPA was added, the degradation solution was light yellow. Roughly 18% of the cotton underwent degradation but no HMF

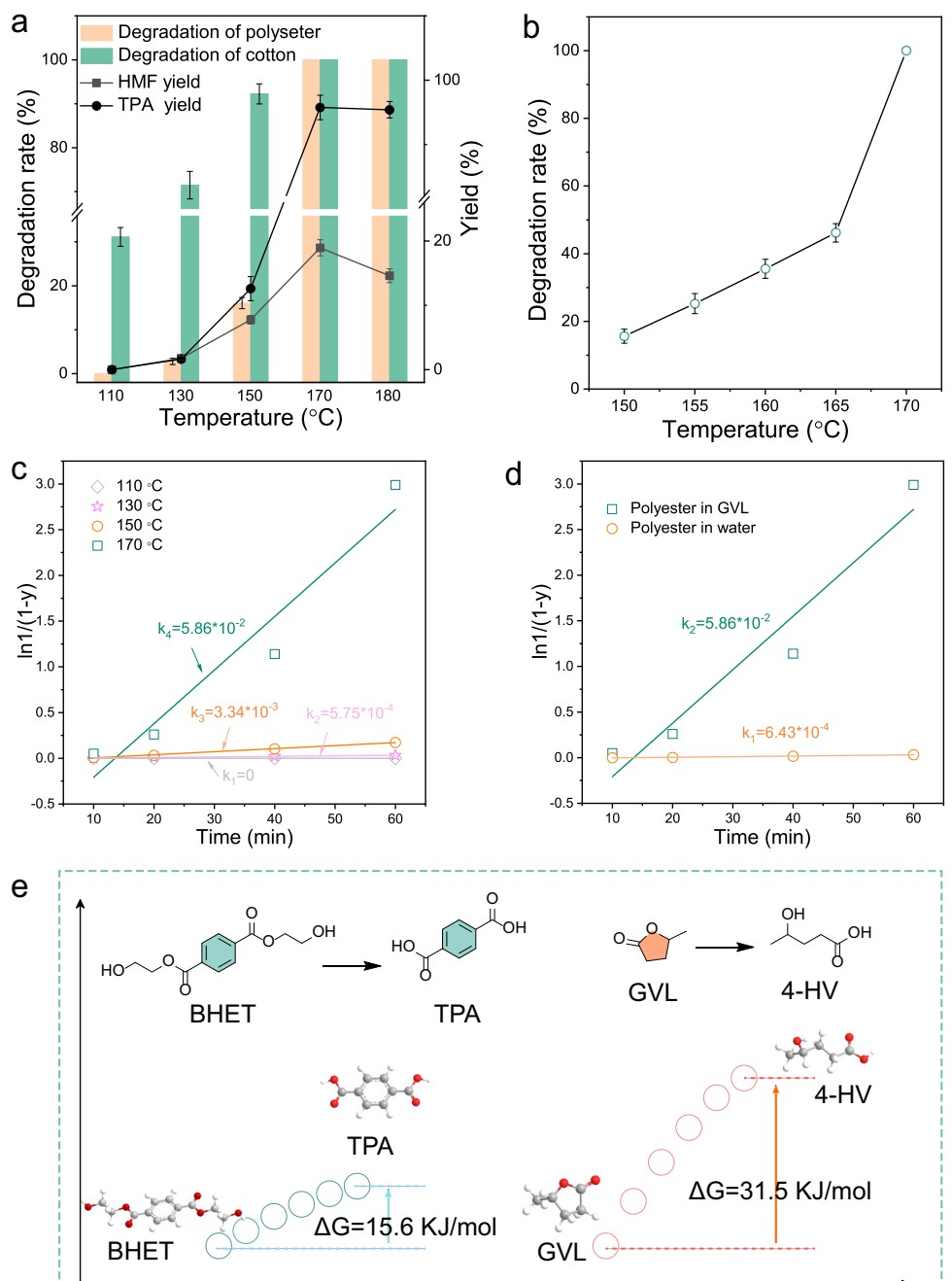

**Fig. 4 | Degradation kinetics studies of polyester/cotton blends. a** The effects of reaction temperature on the degradation of polyester and cotton, and the distribution of products. **b** Variation of polyester degradation rate from 150–170 °C. The data points (**a**) and (**b**) are from N = 3 independent experiments. **c** The reaction rate constants of polyester at different temperatures in GVL. **d** The reaction rate constants of polyester in GVL and aqueous solutions at 170 °C. **e** Calculation of Gibbs free energy of reaction solvent GVL and model compound BHET.

was detected at the presence of TPA (Supplementary Fig. 17), suggesting TPA was only active in catalyzing the degradation of cotton not the conversion to HMF (Fig. 5a). The degradation rate of cotton increased then plateaued as TPA amount increased (Supplementary Fig. 18). This could be attributed to the partial dissolution of TPA in GVL at the reaction temperature, promoting the cotton conversion. It should be noted that EG, the other monomer generated from polyester depolymerization, has negligible impact on the cotton conversion (Supplementary Fig. 19).

Furthermore, the self-separation of TPA from the solvent significantly reduced the acid needed for separation and purification during subsequent processing. It is worth noting that in polyester/cotton blends, the internal water circulation promotes cotton conversion, increasing HMF yield from 19.7% to 25%. HPLC analysis showed that in addition to HMF, the remaining major product was glucose, with a yield of approximately 46%, resulting in an overall carbon yield of 71% (Supplementary Fig. 9). The carbohydrates and HMF can be recovered by extraction from GVL into an aqueous phase by addition of NaCl or liquid $CO_2$[32]. Compared with current work on the recovery of HMF from cotton, our study demonstrated a significant advantage in achieving a higher HMF yield of 25% in GVL (Fig. 5b)[3,27,28,31,33,34]. Even in different blends of polyester and cotton, the HMF yield is significantly higher compared to the one in sole cotton (Fig. 5c). Compared with the traditional hydrolysis method of polyester, a much

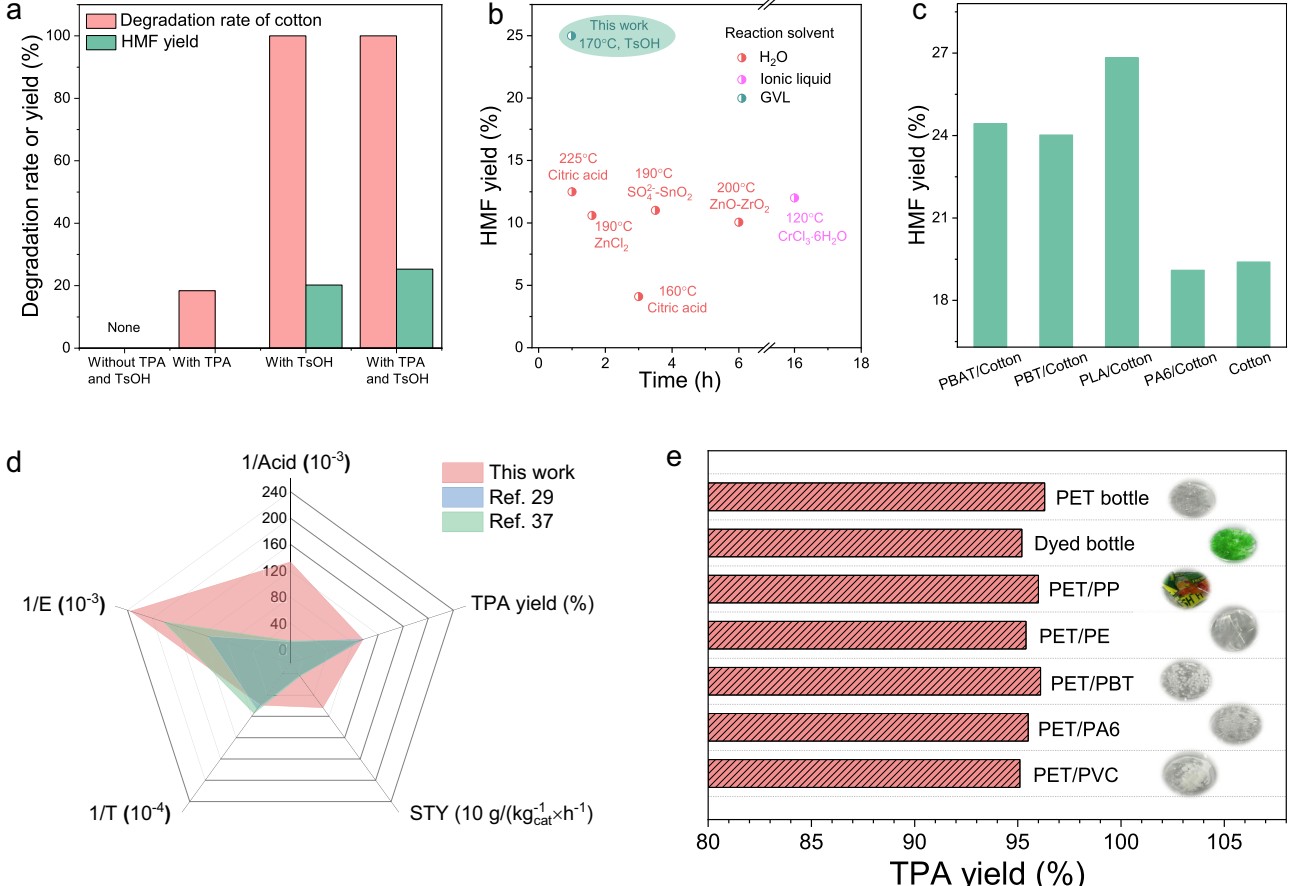

**Fig. 5 | Results of this work compared to traditional work. a** Effects of TPA produced by polyester on cotton conversion in GVL with 7.5 wt% TsOH at 170 °C for 60 min. **b** Comparison of HMF recycled from cotton between this work and different degradation methods in references. **c** Degradation of different cotton mixtures. **d** Radar contrast diagram of this method and traditional hydrolysis method of polyester. **e** Degradation of PET mixed plastics. It should be noted that in the case of PBT/PET mixed plastics, TPA yield refers to the total yield of TPA obtained from the hydrolysis of both PBT and PET. 1/Acid: reciprocal of acid concentration; 1/E: reciprocal of environmental factor; 1/T: reciprocal of reaction temperature; STY: space-time yield.

lower acid concentration was successfully adopted in our work, which is only 10% of that of the traditional method, while obtaining a considerable TPA yield (Fig. 5d)[29,35]. Besides, the whole process of this work exhibits lower environmental factors (E), which may provide valuable references for the recycling process[36]. A lower hydrolysis temperature and a higher space-time conversion rate indicate that the process is energy-efficient. As a further step towards real-world applications, we submitted the system to different PET wastes. The post-consumer PET waste is of high crystallinity and complex mixtures. More than 95% pure white monomer TPA can be obtained by degradation in GVL (Fig. 5e) in various physical mixtures of PET plastics, including polybutylene terephthalate (PBT), polyamide (PA), polyvinylchloride (PVC), polypropylene (PP) and polyethylene (PE). This shows that the method is highly inclusive of impurities and additives in post-consumer PET products and can deal well with commercial PET waste.

## Discussion

In summary, we demonstrate synergistic co-hydrolysis of polyester/ cotton blends in GVL utilizing internal water circulation. Polyester hydrolysis can break the water balance between cotton hydrolysis and subsequently transform into HMF by consuming the water generated during the HMF production step. It drives the reaction in the forward direction of HMF formation, increasing the HMF yield by 25%. The introduction of external water is not conductive to the swelling/dissolution of polyester in GVL and thus decreases degradation rate. This rate constant in GVL is 100 times as much as the one

in an aqueous solution. Besides, the excellent swelling and dissolution properties of GVL on polyester enable the reaction to occur under much lower acid concentrations compared to traditional methods, requiring only one-tenth of the acid concentration used in traditional methods. GVL showed exceptional stability in the reaction. This solidifies the foundation for the cooperative degradation of polyester/cotton blends while effectively preventing excessive degradation of the cotton component. Moreover, HMF yield shows the same upward trend across different polyester-cotton mixture materials. This approach is highly tolerant of impurities and additives in post-consumer PET products and achieves complete PET degradation and high TPA yield in various PET mixed plastics. Considering the diversity of mixed plastics and textiles, we expect that this work will stimulate more development and design of recycling approaches for mixed wastes.

## Methods
### Materials
Polyester/cotton blends (80% Polyester, 20% Cotton), pure polyester textile, and pure cotton were purchased from Jingdong Mall. p-toluenesulfonic acid (TsOH), potassium hydroxide (KOH) and gamma-valerolactone (GVL) were purchased from Chengdu Cologne Chemical Co., Ltd. 5-hydroxymethylfurfural (HMF), glucose and fructose were purchased from Aladdin. Ultra-pure water was used in all experiments. All the above reagents are analytically pure and are not further purified before use.

## Degradation of polyester/cotton blends

Polyester/cotton blends were cut into 3 mm × 3 mm squares after thoroughly dried in vacuum oven at 80 °C. Polyester/cotton blends were added to 50 mL of PTFE-lined hydrothermal reactor with 20 mL GVL solution followed by TsOH. The mass fraction of blends and TsOH were 5% and 7.5%, respectively. Depolymerization was carried out at 170 °C for 60 min with a stirring rate of 400 rpm. Afterward, the precipitate was separated from the solution by filtration to obtain crude terephthalic acid (TPA). An equivalent amount of 1 M KOH aqueous solution was used to dissolve the crude TPA, and then the pH of the solution was adjusted to around 3 using HCl to precipitate TPA and purified TP was finally obtained after filtration.

## Characterization

**Nuclear Magnetic Resonance (NMR) spectra ($^1$H and $^{13}$C).** $^1$H NMR, $^{13}$C NMR spectra were obtained at ambient temperature in DMSO-$d_6$ or $D_2O$ on Bruker Avance III spectrometer operating at 400 MHz. Data for $^1$H NMR and $^{13}$C NMR are recorded as follows: chemical shift (ppm), multiplicity (s, singlet; d, doublet; t, triplet; q, quartet; m, multiplet–splitting patterns that could not be interpreted or easily visualized were designated as multiplet), integration, referenced to the residual solvent peak of DMSO-$d_6$ (2.50 ppm) and $D_2O$ (4.79 ppm). Data for $^{13}$C NMR are reported in terms of chemical shift (ppm) and are referenced to the residual solvent peak of DMSO-$d_6$ (39.6 ppm).

**Fourier Transform Infrared spectra (FTIR).** FT-IR spectra were recorded on a Nicolet 6700 spectrophotometer scanning from 4000 cm$^{-1}$ to 500 cm$^{-1}$. The sample was tableted with potassium bromide in the mass ratio of 1:100.

**X-ray Diffraction (XRD) Analysis.** Samples from different stages of composting (polyester/cotton blends, polyester/spandex blends, recycled cotton, and recycled spandex) were collected, dried, and finely crushed, then analyzed by XRD. XRD was carried out using an X-ray diffractometer (Panalytical EMPYREAN model) with a copper-length anticathode. Operating with CuKα radiation (λ = 1.5406 Å). The current was adjusted to 30 mA and the voltage was increased to 40 kV. Data acquisition is carried out by a control unit for angles of 2 theta (2θ) between 5 and 60°.

**High Efficiency Liquid Chromatography (HPLC).** The concentration of HMF was carried out by HPLC analysis (Shimadzu LC-20AD, Japan) supplied with an Ultimate Plus-C18 column (150 mm × 4.6 mm) at 30 °C. Methanol aqueous (18:82, v:v) solution as the eluent at a flow rate of 1 ml/min. Before the analysis, all the samples were filtered via a 0.22 μm syringe filter and diluted with ultra-pure water. The injection volume was 5 μL per analysis. The concentration of glucose and fructose was carried out by HPLC analysis (Agilent 1260) supplied with an Xtimate Sugar-H column (300 mm × 7.8 mm) at 60 °C. 5 mM sulfuric acid aqueous solution as the eluent at a flow rate of 0.5 ml/min. Before the analysis, all the samples were filtered via a 0.22 μm syringe filter and diluted with ultra-pure water. The injection volume was 10 μL per analysis.

**Karl Fischer titrator.** The water content in the reaction solution was determined using a Karl Fischer titrator (ZDJ-2S, China). Firstly, the titration vessel was cleaned three times with titrant solution, using methanol as the dispersing solvent. Then, pre-titration was carried out until the solution reached a water-free state. 5 μL of pure water was added for calibration, and the water content in the reaction solution was determined after the instrument reached drift equilibrium. Approximately 1 mL of the reaction solution was transferred to the titration vessel for water content determination. Each sample group was tested three times, and the average value was taken.

**Calculation of Gibbs Free Energy.** All calculations were carried out with the Gaussian 16 software. The M06-2X functional[37] and TZVP basis[38] set were adopted for geometry optimization and frequency calculations. The SMD implicit solvation model was used to account for the solvation effect (eps=42.82). Then, the Gibbs free energy change was calculated by the following formula (1):

$$deltaG = G_{product} - G_{reactant} \qquad (1)$$

## Data availability

All data supporting the findings of this study are available within the article, as well as the Supplementary Information file. All data are available from the corresponding author upon request.

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

## Acknowledgements

This work is supported by the National Natural Science Foundation of China (22293063), the Fundamental Research Funds for the Central Universities and Institutional Research Fund from Sichuan University (2020SCUNL205), and the 111 project (B20001).

## Author contributions

Shimei Xu conceived the idea and supervised the whole project. Shun Zhang designed and carried out the experiments. Wenhao Xu, Rong-cheng Du, Lei Yan, and Xuehui Liu discussed the results. Shun Zhang, Shimei Xu, and Yu-Zhong Wang contributed to the writing of the manuscript and commented on the manuscript. All authors approved the final version of the manuscript for submission.

## Competing interests

The authors declare no competing interests.
