## [Peer Review File · Nature Communications]

Internal Water Circulation Mediated Synergistic Co-hydrolysis of PET/cotton Textile Blends in Gamma-valerolactoneReviewers' Comments:

Reviewer #1:

Remarks to the Author:

This is fundamentally important work. The results, demonstrating dual deconstruction of cotton and polyester, have great practical importance. There's enough science value in the approach and the characterization that was performed to merit publication in Nature Communications.

I'm especially impressed with the use of GVL as a solvent and the demonstration that solubility promotes reactivity in textiles. The rate information at 150C (Insolubility) and 170C (Solubility) of 100X is truly dramatic.

My criticisms and suggestions of this important work are two-fold:

1. There are lots of small English mistakes that make reading this rather short manuscript somewhat challenging. Examples are found in the following lines (there's more, these are simply small errors found in the first few pages):

Line 51: "dissolubility"--I don't know what this means.

Line 53: "However, additional...." (not a sentence)

Line 70: "Considering the...."

Line 73-74: "Nevertheless,...."

Line 155: "cotton in the blending"--this 'blending' terminology is consistently used throughout the paper. To me, 'blending' is a verb which means you are actively mixing two dissimilar materials together. I think the authors should change this to something more standard, like "blends" as in cotton/polyester blends. This 'blending' terminology could create confusion for the reader (it confused me initially).

I found these sort of small errors throughout and would suggest one of the editors read the paper carefully in an effort to gain maximum clarity.

2. My other (more important) concern/suggestion is with some of the figures. I find them to be hard to follow and that they create confusion. Most notably, Fig. 3g, which is meant to convey important mechanistic information, only confused me. I think it could be improved fairly easily (or removed). Figure 4e is even more opaque. I gain nothing from staring at it, I think it is important enough for scientific understanding of the author's work that this important figure should be re-done in a way that is clearer.

Lastly: Lines 273-280 where the author's describe complexity in the addition of PA6, I would suggest considering removing that, as it does not add to the paper, simply confuses.

Reviewer #2:

Remarks to the Author:

In this article the authors implement a co-synergetic strategy to facilitate the recycling of PET/cotton blends using low catalyst loading. The main idea of this project was to take advantage from the water formed during the dehydration of fructose to form the HMF as the nucleophile for the PET hydrolysis reaction. According to their claims, the TPA yield was about 95% with a 99% of purity. Additionally, they reported a yield of 25% for HMF, which was superior in comparison to other cotton degradation techniques. The last was a result of the favoured equilibrium towards HMF when water was consumed during the PET hydrolysis. This was proven with several experiments in which in the absence of cotton and just in the solvent and the acid, PET dissolved instead of reacting. Moreover, the same experiment was repeated with cotton but in the absence of PET, the HMF yield was as low as 19.7%.

The current work is interesting and indeed it has the novelties of the proposed synthesis route for TPA

and the conditions for the formation of HMF and should be published as it is highly original and interesting. However, the following points must be clarified in order to consider this work for publication:

- The introduction highlights that cotton/polyester blends are the vast majority of fabric waste. However, in this study, or as it is understandable in the text, the blends treated from both materials are made with pure fabrics in the desired proportions. Did the authors treat real cotton/PET fabric waste (a single piece of fabric made of cotton/polyester yarns)? How would you control the water concentration in this kind of systems? This is an important experiment as maybe additive in the cotton/Polyester can play a role.
- A line or paragraph describing why HMF is valuable or for what it is used will be helpful to highlight the relevance of this new approach.
- Did you try a catalyst different than TsOH? Why did you select this catalyst? Is the key that TsOH is hydrated. The authors should perform this experiment with different catalyst and also if possible with different catalyst concentrations in the selected conditions. TsOH is strong acid if they could manage this reaction with milder acids will be highly beneficial.
- Which method was used to measure the water content in the initial samples and along the reaction? What happens if we take samples that are highly hydrated (or wet). Could the authors perform the experiment with different water contents?
- Which treatment did you give to cotton before starting the experiments? Was it dried? Did you wash the sample previously with another polar solvent to remove the water from the samples?
- Which was the stirring rate used in all the experiments? Was it constant?
- In the sentence from lines 93 to 94, it will be helpful to add the references from the previous works you use as reference.
- Are the reported yields and degradations in mass or molar percentage? Why is not the Relative standard deviation included in the measurements? Did you do the experiments in duplicates? In triplicates?
- How did you know these were the correct ratios?
- In figure 3, you specified the amount and composition of fabric you add for the reaction (solid/liquid ratio). Is this the same amount of solid used in all the reactions? The amount of fabric treated was not mentioned for the previous experiments but the composition of the solid.
- How is this specific solid/liquid ratio chosen?
- Supplementary Fig. 7: it will be appreciated if the figure has a better resolution or it is made larger. The signals are not clear.
- In supplementary table 1: What is the PET to solvent ratio? Where are the Relative Standard Deviation?
- Do the polyesters in the Fig. 5C react in the system of GVL/TsOH? If so, then in the blend PBT/PET did both polyesters react?
- Why do you think the HMF yield was higher in the PLA/cotton blend in comparison to the PET/cotton blend?

- Lines 278 and 279 "Besides, the whole process of this work has a lower environmental factors (E) suggesting a preferred recycling process" How did you reached to this conclusion? I think that if they have an LCA they could include this otherwise it should be removed.

Dear editor:

We would like to thank you for your efficient work in processing our manuscript. We have taken into account all comments made by the reviewers and made careful revisions accordingly. All changes are marked in blue in the manuscript. The point-by-point responses to reviewers are listed below. We hope our revised manuscript can be accepted for publication.

Best regards

Shimei Xu

REVIEWER COMMENTS

Reviewer #1 (Remarks to the Author):

This is fundamentally important work. The results, demonstrating dual deconstruction of cotton and polyester, have great practical importance. There's enough science value in the approach and the characterization that was performed to merit publication in Nature Communications.

I'm especially impressed with the use of GVL as a solvent and the demonstration that solubility promotes reactivity in textiles. The rate information at 150C (Insolubility) and 170C (Solubility) of 100X is truly dramatic.

My criticisms and suggestions of this important work are two-fold:

1. There are lots of small english mistakes that make reading this rather short manuscript somewhat challenging. Examples are found in the following lines (there's more, these are simply small errors found in the first few pages):

Line 51: "dissolubility"--I don't know what this means.

Line 53: "However, additional...." (not a sentence)

Line 70: "Considering the...."

Line 73-74: "Nevertheless,...."

Line 155: "cotton in the blending"--this 'blending' terminology is consistently used throughout the paper. To me, 'blending' is a verb which means you are actively mixing two dissimilar materials together. I think the authors should change this to something more standard, like "blends" as in cotton/polyester blends. This 'blending' terminology could create confusion for the reader (it confused me initially).

I found these sort of small errors throughout and would suggest one of the editors read the paper carefully in an effort to gain maximum clarity.

Response: As suggested, we correct some grammar errors and ambiguous words.

Regarding line 51, the revised content is “these strategies take advantage of the differences in solubility or degradation selectivity among the components of the blends to recover a specific component while preserving the integrity of the others.”

Regarding line 51, the revised content is “However, additional step will be required to recover the other components producing additional energy cost.”

Regarding line 70 and line 73-74, the revised content is “Considering that the acidic hydrolysis of both polyester and cotton is water-consuming process, it is promising to achieve synergistic co-degradation of polyester/cotton blends by utilizing the byproduct water from the HMF dehydration step to meet the water consumption requirement for the hydrolysis of polyester and cotton. However, due to the differences between the two polymers, there are two main issues that need to

be addressed in the acidic co-hydrolysis of polyester/cotton blends into TPA and HMF.” As suggested, we have made the change from “blending” to “blends”.

In addition, the manuscript has been thoroughly revised to polish the English and make the expression scientific.

2. My other (more important) concern/suggestion is with some of the figures. I find them to be hard to follow and that they create confusion. Most notably, Fig. 3g, which is meant to convey important mechanistic information, only confused me. I think it could be improved fairly easily (or removed). Figure 4e is even more opaque. I gain nothing from staring at it, I think it is important enough for scientific understanding of the author's work that this important figure should be re-done in a way that is clearer.

Lastly: Lines 273-280 where the author's describe complexity in the addition of PA6, I would suggest considering removing that, as it does not add to the paper, simply confuses.

Response: Thank you very much for your suggestions. We agree that removing Fig. 3g can effectively avoid confusing the readers. Additionally, we have adjusted Fig. 4e to make it easier to understand. For ease of reference, we have listed the modified Fig. 3 and Fig. 4 below.

Regarding lines 273-280, it is true that the addition of PA6 may make the reader confused since it is not a polyester. In our original manuscript, it was added as a control to confirm the role of polyester hydrolysis. In fact, the role is already confirmed by other different polyesters. As a result, we remove PA6 to make the content clearer as suggested.

Fig. 3 Monitoring the degradation process of polyester/cotton blends. (a) XRD spectra of cotton in GVL and water at different reaction time, (b) XRD spectra of polyester in GVL and water at different reaction time. Reaction conditions: 7.5% TsOH, 170 °C, 20 mL solvent. Polyester treated

by GVL and water at different temperatures for 3 h (c) and different time at 170 °C (d); XRD spectra of polyester (e) and cotton (f) treated with GVL at different time.

Fig. 4 Degradation kinetics studies of polyester/cotton blends. (a) The effects of reaction temperature on the degradation of polyester and cotton, and the distribution of products. (b) Variation of polyester degradation rate from 150-170 °C. (c) The reaction rate constants of polyester at different temperatures in GVL. (d) The reaction rate constants of polyester in GVL and aqueous solutions at 170 °C. (e) Calculation of Gibbs free energy of reaction solvent GVL and model compound BHET.

Reviewer #2 (Remarks to the Author):

In this article the authors implement a co-synergetic strategy to facilitate the recycling of PET/cotton blends using low catalyst loading. The main idea of this project was to take advantage from the water formed during the dehydration of fructose to form the HMF as the nucleophile for the PET hydrolysis reaction. According to their claims, the TPA yield was about 95% with a 99% of purity. Additionally, they reported a yield of 25% for HMF, which was superior in comparison to other cotton degradation techniques. The last was a result of the favoured equilibrium towards HMF when water was consumed during the PET hydrolysis. This was proven with several experiments in which in the absence of cotton and just in the solvent and the acid, PET dissolved instead of reacting. Moreover, the same experiment was repeated with cotton but in the absence of PET, the HMF yield was as low as 19.7%.

The current work is interesting and indeed it has the novelties of the proposed synthesis route for TPA and the conditions for the formation of HMF and should be published as it is highly original and interesting. However, the following points must be clarify in order to consider this work for publication:

- The introduction highlight that the cotton/polyester blends are the vast majority of fabric waste. However, in this study, or as it is understandable in the text, the blends treated from both materials are made with pure fabrics in the desired proportions. Did the authors treated real cotton/PET fabric waste (a single piece of fabric made of cotton/polyester yarns)? How would you control the water concentration in this kind of systems? This is an important experiment as maybe additive in the cotton/Polyester can play a role.

Response: Besides of the blends made with pure fabrics in the desired proportions, we also use commercially available polyester/cotton blends (87% polyester, 13% cotton) to conduct depolymerization experiments (as shown in the following figure I), and successfully achieve 100% polyester depolymerization obtaining HMF. However, for a more in-depth exploration of the conversion mechanism and process, the uncertain proportions of the commercially available blends will complicate the experiment. Therefore, we employed pure fabrics to conduct further research by varying the proportions of the blends freely.

Figure I depolymerization of commercially available polyester/cotton blends into TPA

To control water concentration in this system, we employed the following methods: For the other reactants and fabrics, we thoroughly dried them in an 80 °C vacuum oven before proceeding with the experiments. As for the reaction solvent, after adding TsOH, we utilize a moisture analyzer to measure the water content. For example, the initial water content in the system is measured as 0.31%.

We can maintain the same water content by adding or reducing water.

- A line or paragraph describing why HMF is valuable or for what it is used will be helpful to highlight the relevance of this new approach.

Response: Thank you for the suggestion. We have incorporated descriptions of the applications of HMF into the manuscript. The revised content is “HMF is an important bio-based platform chemical for the production of renewable monomers and bio-fuels, which can be produced by acidolysis of cotton.”

- Did you tried a catalyst different than TsOH? Why did you selected this catalyst? Is the key that TsOH is hydrated. The authors should perform this experiment with different catalyst and also if possible with different catalyst concentrations in the seletled conditions. TsOH is strong acid if they could manage this reaction with milder acids will be highly beneficial.

Response: The selection of TsOH as a catalyst is based on two reasons: On one hand, TsOH possesses acidity comparable to that of inorganic acids, enabling effective catalysis of the depolymerization of cotton-polyester blends, while TsOH exhibits lower equipment corrosion compared to inorganic acids. On the other hand, we previously considered using milder acids for the conversion (such as organic carboxylic acids), but the results were inferior (as shown in the following figure II). PET depolymerization was less efficient.

Figure II Comparison of polyester degradation catalyzed by TsOH, acetic acid (CH₃COOH) and terephthalic acid (TPA). Reaction conditions: 0.8 g polyester, reaction under 170 °C for 60 min with 7.5 wt% catalyst, the GVL is 20 mL.

- Which method was used to measure the water content in the initial samples and along the reaction? What happen if we take samples that are highly hydrated (or wet). Could the authors perform the experiment with different water contents?

Response: In our experiment, the reactants and fabrics were completely dried in an 80 °C vacuum oven before proceeding with the experiments. The thermogravimetric analysis confirmed that there

was no residual moisture in the sample (Figure II). After adding TsOH, we utilize a moisture analyzer to measure the water content of the system using a Karl Fischer titrator (ZDJ-2S, China). The water content during the reaction process was also determined by the method. Firstly, the titration vessel was cleaned three times with titrant solution, using methanol as the dispersing solvent. Then, pre-titration was carried out until the solution reached a water-free state. 5 μL of pure water were added for calibration, and the water content in the reaction solution was determined after the instrument reached drift equilibrium. Approximately 1 mL of the reaction solution was transferred to the titration vessel for water content determination. Each sample group was tested three times, and the average value was taken. Furthermore, we have added the relevant content in the supplementary information.

In Fig. 2c and Fig. 2d, we investigated the effects of different water contents on polyester and cotton. We found that when the water content is below 5%, the depolymerization of polyester is essentially unaffected, but the yield of HMF decreases with increasing water content. This is because the increased water content is unfavorable for the generation of HMF. Therefore, conducting experiments in the laboratory with highly hydrated or moist samples will result in a decrease in the yield of low HMF, while the depolymerization rate of polyester remains essentially unchanged. To avoid the decrease in yield, we will remove the water by evaporation to maintain the same water content.

Fig. 2 Mechanism of internal circulation of water. (c) Effect of different contents of water in GVL on HMF yield from cotton and (d) on the degradation rate of polyester.

• Which treatment did you give to cotton before starting the experiments? Was it dried? Did you washed the sample previously with another polar solvent to remove the water from the samples?

Response: Before the experiment begins, we thoroughly dry the cotton in a vacuum oven at 80°C. And we did not wash the sample previously with another polar solvent. To demonstrate the effectiveness of drying, we conducted thermogravimetric analysis on the thoroughly dried samples. As shown in the Figure III below, there is almost no weight loss observed in the samples before 100°C, indicating that the water content in the samples has been adequately removed.

Figure III. The thermogravimetric curves of polyester (a) and cotton (b) after thorough drying.

- Which was the stirring rate used in all the experiments? Was it constant?

Response: All experiments were conducted at a constant stirring speed of 400 rpm.

- In the sentence from lines 93 to 94, it will be helpful to add the references from the previous works you use as reference.

Response: As suggested, we added the reference of the previous works as reference 30.

- Are the reported yields and degradations in mass or molar percentage? Why is not the Relative standard deviation included in the measurements? Did you do the experiments in duplicates? In triplicates?

Response: In our work, we calculate yields and degradation rates based on mass percentages. Additionally, we have conducted supplementary experiments to determine the relative standard deviations.

Table1. Degradation of polyester and cotton. ^[a]

Entry	Substrate	Solvent	Yield (%)	
			TPA ^[c]	HMF ^[d]
1	polyester	GVL	96.3±1.6	/
2	polyester	H ₂ O	< 1	/
3	cotton	GVL	/	19.7±1.1
4	cotton	H ₂ O	/	4.34±1.1
5	polyester/cotton	GVL	95.8±1.9	24.1±1.0
6	polyester/cotton	H ₂ O	< 1	4.73±0.9

Polyester and cotton were cut from textiles; [a] Reaction under 170 °C for 60 min with 7.5 wt% TsOH, the solvent is 20 mL. Unless otherwise specified, the polyester/cotton blends utilized in this work is 1 g (with a polyester to cotton mass ratio of 8:2). In experiments involving the individual use of cotton or polyester, the masses of polyester and cotton are 0.8 g and 0.2 g, respectively. [b] The volume ratio of GVL to H₂O is 5:5; [c] Yield determined by weight; [d] Yield determined by NMR and HPLC.

Fig. 2 Mechanism of internal circulation of water. (a) The effects of polyester/cotton blends ratios on HMF yield distribution

Fig. 4 Degradation kinetics studies of polyester/cotton blends. (a) The effects of reaction temperature on the degradation of polyester and cotton, and the distribution of products. (b) Variation of polyester degradation rate from 150-170 °C

• How did you know these were the correct ratios?

Response: For textile blends, composition ratios can be determined by solid-state nuclear magnetic resonance (solid-state NMR) or by dissolving one of its components using a solvent to ascertain its

composition ratio.

For solid/liquid ratio, when using 0.8 g of polyester for the experiment, it was observed that a low volume of solvent cannot immerse completely the polyester. However, considering that the polyester would undergo swelling, and the remaining solution might fail to completely immerse the PET, resulting in a deviation in mass transfer. When the solvent volume is 20 mL, it effectively swells and dissolves the polyester (as shown in the Figure IV and Fig. 3c-d). Further increasing the solvent amount would result in reduction of the reactant concentration, which is not conducive to the progress of the reaction and would also increase economic costs. Therefore, choosing a solvent volume of 20 mL is more appropriate.

Figure IV Comparison of different solid-to-liquid ratios. The state of 0.8 g of polyester dissolved in 5 mL, 10 mL, and 20 mL of GVL respectively. The black dashed line indicates the position of the liquid level.

Fig.3 Polyester treated by GVL and water at different temperatures for 3 h (c) and different time at 170 °C (d).

• In figure 3, you specified the amount and composition of fabric you add for the reaction (solid/liquid ratio). Is this the same amount of solid used in all the reactions? The amount of fabric treated was not mentioned for the previous experiments but the composition of the solid.

Response: In all reactions, unless otherwise specified, the mass of the polyester/cotton blends used is 1 g (with a polyester to cotton mass ratio of 8:2). When conducting experiments using polyester or cotton separately, the masses of polyester and cotton are 0.8 g and 0.2 g, respectively. To facilitate

a clearer understanding of the experimental details for readers, we have provided detailed explanations above in Table 1.

- How is this specific solid/liquid ratio chosen?

Response: When using 0.8 g of polyester for the experiment, it was observed that a low solvent volume (5 mL) cannot immerse completely the polyester. When the solution volume was exactly 10 mL, the liquid level just submerged the polyester (as shown in the Figure IV below). However, considering that the polyester would undergo swelling, and the remaining solution might fail to completely immerse the PET, resulting in a deviation in mass transfer. Additionally, if a polyester/cotton blends is used, it would necessitate a further increase in the amount of GVL used. To ensure sufficient contact between the solvent and substrate during the reaction process, we increased the solvent volume to 20 mL.

Figure IV Comparison of different solid-to-liquid ratios. The state of 0.8 g of polyester dissolved in 5 mL, 10 mL, and 20 mL of GVL respectively. The black dashed line indicates the position of the liquid level.

- Supplementary Fig. 7: it will be appreciated if the figure has a better resolution or it is make larger. The signals are not clear.

Response: "As suggested, we have made modifications to Supplementary Fig. 7 for better clarity of the signals. The following image is the revised version.

Supplementary Fig. 7 NMR spectra of reaction solution derived from polyester/cotton blending with different time in water (A). NMR spectra with local magnification (B).

• In supplementary table 1: What is the PET to solvent ratio? Where are the Relative Standard Deviation?

Response: We have added the quantities of polymers and their component ratios to supplementary Table 1. The modified table is as shown below. As you can see from the Table 1 and Fig. 2a and Fig. 4a and 4b, the experiments exhibit satisfactory repeatability with small relative standard deviation. Table S1 represents the product change over time, it is clear enough to reveal variations in the pattern. Therefore, the Relative Standard Deviation is not given since it does not affect our conclusions in this case.

Table1. Degradation of polyester and cotton. ^[a]

Entry	Substrate	Solvent	Yield (%)	
			TPA ^[c]	HMF ^[d]
1	polyester	GVL	96.3±1.6	/
2	polyester	H ₂ O	< 1	/
3	cotton	GVL	/	19.7±1.1
4	cotton	H ₂ O	/	4.34±1.1
5	polyester/cotton	GVL	95.8±1.9	24.1±1.0
6	polyester/cotton	H ₂ O	< 1	4.73±0.9
7	polyester/cotton	GVL/ H ₂ O ^[b]	28.3±1.4	7.38±1.3

Polyester and cotton were cut from textiles; [a] Reaction under 170 °C for 60 min with 7.5 wt% TsOH, the solvent is 20 mL. Unless otherwise specified, the polyester/cotton blends utilized in this work is 1 g (with a polyester to cotton mass ratio of 8:2). In experiments involving the individual use of cotton or polyester, the masses of polyester and cotton are 0.8 g and 0.2 g, respectively. [b] The volume ratio of GVL to H₂O is 5:5; [c] Yield determined by weight; [d] Yield determined by NMR and HPLC.

Fig. 2 Mechanism of internal circulation of water. (a) The effects of polyester/cotton blends ratios on HMF yield distribution

Fig. 4 Degradation kinetics studies of polyester/cotton blends. (a) The effects of reaction temperature on the degradation of polyester and cotton, and the distribution of products. (b) Variation of polyester degradation rate from 150-170 °C

Supplementary Table 1. Depolymerization of polyester/cotton blends and HMF yield. ^[a]

Entry	Time (min)	5-HMF yield (%) ^[b]	Degradation rate (%)	
			Cotton	Polyester
1	10	0	6.2	0
2	20	1.9	21.2	8.6
3	30	13.5	52.8	23.5
4	40	20.7	97.8	47.8
5	60	24.8	100	100

Polyester and cotton were cut from textiles; [a] 1 g polyester/cotton blends (with a polyester to cotton mass ratio of 8:2), reaction under 170 °C for 60 min with 7.5 wt% TsOH, the GVL is 20 mL; [b] Yield determined by NMR and HPLC.

- Do the polyesters in the Fig. 5C react in the system of GVL/TsOH? If so, then in the blend PBT/PET did both polyesters react?

Response: The polyester materials (PBAT, PBT, PLA) mentioned in Figure 5C undergo depolymerization in the system. It is precisely because the depolymerization of polyester promotes the internal circulation of water, leading to higher yields of HMF compared to pure cotton. In the case of a PET and PBT blend, due to the lower activation energy of reaction for PBT (Polymer-Plastics Technology and Engineering, 45, 171–181, 2004) compared to PET (Polymer-Plastics Technology and Engineering, 43, 1093–1113, 2006), both PET and PBT undergo depolymerization under the reaction conditions. In this case, TPA yield refers to the total yield of TPA obtained from the hydrolysis of both PBT and PET. We have provided relevant explanations in Fig. 5e.

Fig. 5 Results of this work compared to traditional work. (e) Degradation of PET mixed plastics. It should be noted that in the case of PBT/PET mixed plastics, TPA yield refers to the total yield of TPA obtained from the hydrolysis of both PBT and PET.

- Why do you think the HMF yield was higher in the PLA/cotton blend in comparison to the PET/cotton blend?

Response: In our experiments on the depolymerization of different polyester/cotton blends, we added the same amount of polyester. However, due to the higher molecular weight of the structural units of PET (192 g.mol⁻¹) compared to PLA (78.08 g.mol⁻¹), at the same mass, PLA contains relatively more ester functional groups. This can further promote water cycling, resulting in a slightly higher yield of HMF than PET.

- Lines 278 and 279 “Besides, the whole process of this work has a lower environmental factors (E) suggesting a preferred recycling process” How did you reached to this conclusion? I think that if they have an LCA they could include this otherwise it should be removed.

Response: Thank you very much for pointing out the issues. It is indeed not appropriate to solely use low environmental factors to illustrate this as a preferred recycling process. Low environmental factors do facilitate the recycling process. Therefore, we have made some adjustments to the statement, the revised content is “Besides, the whole process of this work exhibits a lower environmental factors (E), which may provide valuable references for the recycling process.”

Reviewers' Comments:

Reviewer #1:

Remarks to the Author:

Thanks for taking the review comments seriously. I think you addressed them well (both Rev. 1 and 2) and think the paper is now in good shape and should be published.

Reviewer #2:

Remarks to the Author:

The authors have adressed all my comments and I am happy with this version.